# Identification and Characterization of Cancer-Associated Fibroblast Subpopulations in Lung Adenocarcinoma

**DOI:** 10.3390/cancers14143486

**Published:** 2022-07-18

**Authors:** Daeseung Kim, Jeong Seon Kim, Inyoung Cheon, Seo Ree Kim, Sang Hoon Chun, Jae Jun Kim, Sieun Lee, Jung Sook Yoon, Soon Auck Hong, Hye Sung Won, Keunsoo Kang, Young-Ho Ahn, Yoon Ho Ko

**Affiliations:** 1Deargen Inc., Daejeon 34051, Korea; daeseung.kim@deargen.me; 2Department of Molecular Medicine and Inflammation-Cancer Microenvironment Research Center, College of Medicine, Ewha Womans University, Seoul 07804, Korea; jeongseonkim821@gmail.com (J.S.K.); 5901ciy@naver.com (I.C.); hellosieun@gmail.com (S.L.); 3Department of Molecular and Cellular Biochemistry, University of Kentucky, Lexington, KY 40536, USA; 4Department of Internal Medicine, Division of Oncology, College of Medicine, The Catholic University of Korea, Seoul 06591, Korea; seoreek@gmail.com (S.R.K.); rowett@catholic.ac.kr (S.H.C.); woncomet@catholic.ac.kr (H.S.W.); 5Department of Thoracic and Cardiovascular Surgery, Uijeongbu St. Mary’s Hospital, College of Medicine, The Catholic University of Korea, Seoul 06591, Korea; medkjj@catholic.ac.kr; 6Uijeongbu St. Mary’s Hospital Clinical Research Laboratory, The Catholic University of Korea, Uijeongbu 11765, Korea; ibbissbb@hanmail.net; 7Department of Pathology, College of Medicine, Chung-Ang University, Seoul 06974, Korea; hsu108@hanmail.net; 8Department of Microbiology, College of Science & Technology, Dankook University, Cheonan 31116, Korea; kangk1204@gmail.com; 9Cancer Research Institute, College of Medicine, The Catholic University of Korea, Seoul 06591, Korea

**Keywords:** cancer-associated fibroblasts, single-cell RNA sequencing, lung adenocarcinoma, tumor microenvironment, heterogeneity

## Abstract

**Simple Summary:**

Considering that cancer-associated fibroblasts (CAFs) facilitate cancer cell motility, invasiveness, and drug resistance, heterogeneity within the CAF population may be of major significance during cancer progression and metastasis. Through pseudotime trajectory analysis of single-cell RNA sequencing data, we revealed several subpopulations of lung CAFs with distinct gene expression patterns, namely, immunosuppressive, neoantigen-presenting, myofibroblastic, and proliferative CAFs. The knockdown of *KPNA2*, one of the neoantigen-presenting CAF-specific markers, attenuated CAF invasiveness, suggesting that CAF subtype markers may represent therapeutic targets within the tumor microenvironment of lung cancer patients.

**Abstract:**

Cancer-associated fibroblasts (CAFs) reside within the tumor microenvironment, facilitating cancer progression and metastasis via direct and indirect interactions with cancer cells and other stromal cell types. CAFs are composed of heterogeneous subpopulations of activated fibroblasts, including myofibroblastic, inflammatory, and immunosuppressive CAFs. In this study, we sought to identify subpopulations of CAFs isolated from human lung adenocarcinomas and describe their transcriptomic and functional characteristics through single-cell RNA sequencing (scRNA-seq) and subsequent bioinformatics analyses. Cell trajectory analysis of combined total and THY1 + CAFs revealed two branching points with five distinct branches. Based on Gene Ontology analysis, we denoted Branch 1 as “immunosuppressive”, Branch 2 as “neoantigen presenting”, Branch 4 as “myofibroblastic”, and Branch 5 as “proliferative” CAFs. We selected representative branch-specific markers and measured their expression levels in total and THY1 + CAFs. We also investigated the effects of these markers on CAF activity under coculture with lung cancer cells. This study describes novel subpopulations of CAFs in lung adenocarcinoma, highlighting their potential value as therapeutic targets.

## 1. Introduction

As epithelial tumors progress, surrounding stromal tissues undergo extensive remodeling under the influence of various growth factors and cytokines, changes in the extracellular matrix, the recruitment of diverse cell types (e.g., fibroblasts, immune cells, pericytes, and vascular endothelial cells), and the formation of new blood vessels [1,2]. As the active tumor stroma provides an optimal environment for cancer cell proliferation, migration, and invasion, the regulation of dynamic changes within stromal compartments is a crucial determinant of cancer progression [3]. Cancer-associated fibroblasts (CAFs) are the main cellular components of the tumor stroma, facilitating cancer cell migration and invasion via direct and indirect interactions [4,5,6]. Through such bidirectional signaling, CAFs and cancer cells stimulate each other to promote cancer progression and metastasis [6,7,8].

For decades, lung cancer has remained the leading cause of cancer-associated death worldwide [9]. Although innovative therapies, such as targeted therapy and immunotherapy, have been introduced, the prognosis of lung cancer patients remains suboptimal, which is mainly attributed to tumor heterogeneity and treatment resistance [10,11]. During cancer development and progression, tumor cells acquire greater genetic and phenotypic diversity, thus becoming more heterogeneous in terms of gene expression, morphology, proliferation, motility, invasiveness, and metastatic potential [12,13]. Such tumor heterogeneity is considered to cause resistance to chemotherapy. In addition, a growing body of evidence supports the notion that not only cancer cells but also other cell types within the tumor microenvironment, such as macrophages [14] and CAFs [15,16,17,18,19], exhibit considerable heterogeneity. As CAFs are known to facilitate cancer cell motility, invasiveness, and drug resistance, CAF heterogeneity may be a major determinant of cancer progression and metastasis.

Cellular heterogeneity can be explored through single-cell RNA sequencing (scRNA-seq) [20]. After securing transcriptomic data of individual cells using a molecular barcode system, data dimensions are reduced based on informative selected features, followed by clustering and trajectory inference to dissect cellular heterogeneity. ScRNA-seq analysis of breast CAFs previously revealed functionally distinct CAF subpopulations, including vascular, matrix, cycling, and developmental CAFs [15]. In pancreatic ductal adenocarcinoma, myofibroblastic, inflammatory, and antigen-presenting CAFs were identified [21]. CAFs from primary and metastatic head and neck tumors were divided into four subtypes, namely, myofibroblasts, nonactivated resting fibroblasts, and two activated CAF subpopulations (CAF1 and CAF2) [22]. Within the lung tumor microenvironment, five distinct fibroblast types were identified, which exhibit distinct expression patterns with regard to collagen and other extracellular matrix (ECM) components [23].

Most scRNA-seq studies have used whole tumor tissues and then extracted the CAF population via bioinformatics analysis. For a more accurate investigation of CAF subpopulations in lung adenocarcinoma, we isolated CAFs from human lung tumors based on cell-type-specific markers prior to performing scRNA-seq. Within the whole CAF population, clustering analyses and trajectory inference revealed several subpopulations with distinct gene expression patterns. Furthermore, knockdown of *KPNA2*, a Branch 2-specific marker, attenuated CAF invasiveness, suggesting that lung CAFs can be divided into functionally distinct subpopulations.

## 2. Materials and Methods

### 2.1. Isolation of CAFs

This study was approved by the Institutional Review Board of the Catholic Medical Center (No. UC21EISI0118) and was performed as per guidelines for human research. Only patients with untreated, primary, nonmetastatic lung tumors that underwent lung lobe resection with curative intent and provided informed consent were included in this study. Following surgical resection, tumor samples were isolated and immediately transported to the research facility. Tumor samples were rinsed with PBS and soaked overnight in tubes containing 2 mL MACS Tissue Storage Solution (Miltenyi Biotec, Bergisch Gladbach, Germany). CAFs were isolated from the tumors via magnetic-activated cell sorting with antibodies against cell-type-specific markers as previously described [24]. Tumors were dissociated into single cells by treatment with collagenase and Dispase II (3 μg/mL, Thermo Fisher Scientific, Waltham, MA, USA) on a gentleMACS Dissociator (Miltenyi Biotec). Total CAFs were obtained after negative selection based on other cell type markers, including CD45 (lymphocytes; BD #555480, Franklin Lakes, NJ, USA), CD68 (macrophages; BD #556059), CD31 (endothelial cells; BD #550389), and Ep-CAM (epithelial cells; BD #554276). THY1 + CAFs were then isolated via positive selection based on THY1 (fibroblasts; BD #555593). CAFs were cultured in alpha-MEM (Welgene, Gyeongsan, Korea) containing 10% fetal bovine serum (FBS; HyClone, Logan, UT, USA), penicillin/streptomycin (100 U/mL and 100 μg/mL, Welgene), 2 mM L-glutamine (Welgene), and 1 mM sodium pyruvate (Welgene) at 37 °C and 5% CO_2_.

### 2.2. Cell Culture

344SQ murine lung cancer cells and A549 and HCC827 human lung cancer cells were cultured in RPMI 1640 (Welgene) with 10% FBS. Raw264.7 murine macrophages were cultured in DMEM (Welgene) with 10% FBS. Normal murine lung fibroblasts (LFs) and CAFs were isolated and cultured as previously described [24]. *Ube2t*, *Tk1*, *Cxcl12*, and *Kpna2* siRNAs were purchased from Bioneer (Daejeon, Korea) and transiently transfected into murine CAFs using a TransIT-X2 Dynamic Delivery System (Mirus Bio, Madison, WI, USA). Mouse *Kpna2* cDNA (pCMV-SPORT6_*Kpna2*, # mMU001935) was purchased from the Korea Human Gene Bank, Medical Genomics Research Center, KRIBB, Korea, and inserted into a pLVX-Blast vector, which was modified from the pLVX-Puro vector (Clontech, Mountain View, CA, USA). Mouse *Kpna2* was introduced into LFs via lentiviral infection [5]. Human *KPNA2* shRNAs were from Sigma-Aldrich (St. Louis, MO, USA) and introduced into A549 and HCC827 cells via lentiviral infection. Spheroid invasion assay with 344SQ (labeled with mCherry) and murine fibroblasts (labeled with GFP) and Transwell migration assay were performed as previously described [5]. For anticancer drug tests, fluorescence-labeled 344SQ cells (red, 5 × 10^4^) and LFs (green, 1 × 10^4^) were cocultured in 24-well plates and were treated with cisplatin and 5-fluorouracil for 2 days. Six random microscopic fields were photographed under a fluorescence microscope, and the areas covered by 344SQ cells were measured by ImageJ (NIH, Bethesda, MD, USA).

### 2.3. Quantitative Reverse Transcription-PCR (qRT-PCR)

Total RNA was isolated from CAFs or human lung tumors using an AccuPrep Universal RNA Extraction Kit (Bioneer) according to the manufacturer’s protocol. After reverse transcription with an Elpis RT Prime Kit (Elpis Biotech, Daejeon, Korea), real-time PCR was performed using a BioFACT A-Star Real-Time PCR Kit with SFCgreen I (BioFACT, Daejeon, Korea). Target gene mRNA levels were normalized to those of *Rpl32* mRNA. The qRT-PCR primers used in this paper are listed in Appendix A.

### 2.4. Single-Cell RNA Sequencing and Data Analysis

Cells were loaded onto a 10X Chromium single-cell encapsulation chip according to the manufacturer’s instructions (10X Genomics, Pleasanton, CA, USA). Libraries were prepared with a Chromium Single Cell 3′ v2 Reagent Kit. Sequencing was performed on Illumina HiSeq 2500 (San Diego, CA, USA). Upon completion of sequencing, the raw files were processed using a 10X Genomics Cell Ranger (version 3.0.2) toolkit. All quality-passed reads were mapped to the reference human genome (GRCh37). Seurat (version 3.1.2) [25] was used to quantify the expression levels of known genes in single cells. To filter out low-quality and/or dying cells and doublets, cells that had over 2500 unique feature counts were discarded. Cells that had >10% mitochondrial gene counts were also removed. After removing unreliable cells, the unique counts of all genes were log-transformed, and all sample data were integrated into one object using the IntegrateData function. For a proper comparison, all feature counts were individually regressed against each feature, and the resulting residuals were then scaled and centered using the ScaleData function. All downstream analyses were conducted with these processed data.

### 2.5. Bulk mRNA Sequencing and Data Analysis

Libraries were constructed using a SureSelect Strand-Specific Library Prep kit and RNA extracted from total and THY1 + CAFs and then sequenced using Illumina HiSeq 2500. Sequenced reads were trimmed according to sequencing quality using Trimmomatic [26] and then aligned to the reference human genome (GRCh37) using STAR [27]. RSEM [28] was used to estimate the abundance of all known genes as per the transcripts per kilobase million (TPM) method. CAF-associated genes were defined if the average of TPM values across samples was greater than 1 and the fold change was greater or less than 2.

### 2.6. Unsupervised Clustering

The identified CAF-associated genes were used as variable feature genes for the unsupervised dimensional reduction analysis using Seurat [25]. Primary components were selected using the JackStraw and PCEIbowPlot functions implemented in Seurat. Unsupervised clustering was conducted using the FindClusters function with the primary components, and clusters were visualized on uniform manifold approximation and projection (UMAP) and t-distributed stochastic neighbor embedding (t-SNE) plots.

### 2.7. CAF Subtype Prediction via Trajectory Analysis

Cell trajectory and dimensional reduction analysis were performed using Monocle2 [29] with the identified CAF-associated genes. To determine CAF subtype-specific biomarker genes (signature genes), differential gene expression analysis was conducted using the FindAllMarkers function implemented in the Seurat package. Subtype-specific signature genes were defined if the following criteria were met: log2 of fold change was greater or less than 0.15, adjusted *p*-value was less than or equal to 0.05, and expression was detected in at least 30% of cells. Gene Ontology (GO) analysis was conducted using g:Profiler [30] with the signature genes. The functional inference of trajectory-based CAF subtypes was conducted according to GO analysis. A deep-learning-based classification algorithm, called Wx [31], was used to identify reliable biomarker genes that can accurately distinguish CAF subtypes.

### 2.8. Survival Analysis

Total RNAs isolated from surgically resected human lung adenocarcinoma tissues (*n* = 48) were subjected to qRT-PCR as described above to measure *UBE2T*, *TK1*, *CXCL12*, and *KPNA2* mRNA levels. Patients were divided into two groups (high and low) based on their mRNA levels. The optimal cutoff with the most significant (log-rank test) split was determined using the survminer R package, and Kaplan–Meier analysis of patient disease-free survival was performed using GraphPad Prism software.

### 2.9. Mouse Experiments

Before initiation, all the proposed mouse studies were approved by the Institutional Animal Care and Use Committee (IACUC) of Ewha Womans University College of Medicine (EUM20-063). Mice were cared for and euthanized according to standards set forth by the IACUC. Syngeneic (129/Sv) mice were orthotopically injected in the left lung with 344SQ (5 × 10^4^ cells) and LFs (5 × 10^4^ cells per mouse). After 11 days, the mice were necropsied to determine primary and metastatic tumor growth.

### 2.10. Immunohistochemistry (IHC)

Paraffin block-embedded tumor sections were subjected to IHC. Briefly, the sections were deparaffinized and rehydrated using xylene and alcohol. Antigen retrieval was performed by heating the slides for 20 min in Tris-EDTA buffer (pH 9.0), and the endogenous peroxidase activity was blocked by quenching with 3% hydrogen peroxide in methanol for 10 min. The following steps were performed using ImmPRESS HRP Universal Antibody (Horse Anti-Mouse/Rabbit IgG) Polymer Detection Kit, Peroxidase (MP-7500; Vector Laboratories, Burlingame, CA, USA), according to the manufacturer’s instructions. The sections were incubated with primary antibodies against rabbit polyclonal KPNA2 (1:200, ab84440; Abcam, Cambridge, UK) overnight at 4 °C. Antibody binding was visualized with an ImmPACT DAB Peroxidase (HRP) Substrate kit (SK-4105; Vector Laboratories).

### 2.11. Statistical Analysis

Data were analyzed with the Student’s *t*-test and log-rank test using GraphPad Prism (La Jolla, CA, USA) unless otherwise noted.

## 3. Results

### 3.1. Isolation of CAFs from Human Lung Adenocarcinoma

We employed magnetic-activated cell sorting to isolate CAFs from a tumor surgically resected from a patient diagnosed with lung adenocarcinoma (mutant EGFR; low PD-L1). The tumor mass was enzymatically and mechanically digested into single cells, and other cell types, such as lymphocytes, macrophages, vascular endothelial cells, and epithelial cancer cells, were removed through negative selection based on cell-type-specific markers (CD45, CD68, CD31, and Ep-CAM, respectively). The cells contained in this eluate were named “total CAFs”. Total CAFs were again subjected to positive selection for a CAF-specific marker (THY1), and enriched cells were named “THY1 + CAFs” (Figure 1A).

Morphologically, both total CAFs and THY1 + CAFs retained active fibroblast characteristics, including elongated spindle or stellate shapes, long thin cytoplasmic projections, and prominent actin stress fibers (Figure 1B). In these CAFs, the mRNA expression levels of marker genes were analyzed via quantitative RT-PCR (qRT-PCR). *THY1* mRNA was highly expressed in both CAF populations, while little to no *CD45*, *CD31*, and *EpCAM* expression was observed. However, CAFs exhibited significant *CD68* mRNA expression (Figure 1C). It should be noted that the detection of cytosolic *CD68* mRNA does not indicate its cell surface expression. Further, CD68 is known to be expressed in nonmyeloid cells, including fibroblasts [32]. CAF markers exhibited high expression in both total and THY1 + CAFs (Figure 1D), confirming successful and specific CAF isolation from the lung tumor.

### 3.2. Single-Cell RNA Sequencing (scRNA-seq) of Human Lung CAFs

A growing body of evidence supports the heterogeneity of CAFs within the tumor microenvironment of epithelial cancers, including pancreatic ductal adenocarcinoma [19], breast cancer [18], and lung adenocarcinoma [23]. To further elucidate the functional heterogeneity of lung CAFs and to identify specific markers of CAF subpopulations, we subjected total and THY1 + CAFs to 10X Genomics scRNA-seq. After passing quality control, two independent scRNA-seq datasets from 5139 total CAFs (mean reads per cell = 122,073; median genes per cell = 4371) and 4038 THY1 + CAFs (mean reads per cell = 138,550; median genes per cell = 5076) were integrated and directed to subsequent analysis (Figure 2A). To characterize expression differences between total and THY1 + CAFs, differentially expressed genes (DEGs) obtained via additional bulk RNA-seq of the two CAF types were adopted as the main feature genes for clustering analyses. Unsupervised clustering analyses employing uniform manifold approximation and projection (UMAP; Figure 2B) or t-distributed stochastic neighbor embedding (t-SNE; Figure 2C) algorithms subdivided the CAF population into eight subpopulations (Clusters 0–7). When focusing on the expression levels of cluster-specific marker genes, a clear distinction was observed between each cluster (Figure 2D). Among enriched GO terms were mitotic cell cycle (Cluster 0), protein localization (Clusters 1 and 5), mRNA process (Cluster 2), DNA/RNA metabolic process (Cluster 3), ECM and cell motility (Cluster 4), apoptotic process (Cluster 6), and metabolic process (Cluster 7) (Figure 2E). Similarly to the qRT-PCR results in Figure 1C, *CD45*, *EpCAM*, and *CD31* exhibited little or no expression, *CD68* showed intermediate expression, and *THY1* was highly expressed in all clusters (Figure 2F,G). These results suggest that diverse subpopulations of CAFs exist even within a single tumor from a lung cancer patient, and these CAFs can be distinguished via scRNA-seq analysis.

### 3.3. Cell Trajectory Analysis of CAF scRNA-seq Data

Next, we employed pseudotime trajectory analysis in order to identify different cell lineages or subpopulations based on CAF scRNA-seq data [33]. Trajectory analysis revealed two branching points and five distinct branches (Figure 3A). Interestingly, total and THY1 + CAFs were segregated, with the latter being dominantly enriched in Branch 1 (51.9% of THY1 + CAFs) and Branch 2 (45.3%), while the former were dominantly enriched in Branch 4 (19.8% of total CAFs) and Branch 5 (77.2%; Figure 3B,D). We then extracted DEGs between each branch and designated these genes as branch-specific markers. Based on GO term analysis of these markers, each branch was functionally denominated (Figure 3C,D). Branch 1 markers were enriched in GO terms, including “regulation of apoptotic process (*p* = 1.5 × 10^−^^12^)”, “response to cytokine (*p* = 7.3 × 10^−^^11^)”, and “immune system process (*p* = 0.008)”. Therefore, we named Branch 1 “immunosuppressive CAFs”. Branch 2 markers were associated with “cellular response to cytokine stimulus (*p* = 1.4 × 10^−^^13^)”, “antigen processing and presentation (*p* = 1.7 × 10^−^^13^)”, and “T cell receptor signaling pathway (*p* = 3.9 × 10^−^^6^)”. Therefore, Branch 2 CAFs were named “neoantigen-presenting CAFs”. Branch 4 was termed “myofibroblastic CAFs” because its markers were enriched in “extracellular matrix organization (*p* = 6.9 × 10^−^^19^)”, “cell migration (*p* = 2.6 × 10^−^^9^)”, and “cell–matrix adhesion (*p* = 5.6 × 10^−^^8^)”. Branch 5 was named “proliferative CAFs” as its markers were related to “mitotic cell cycle process (*p* = 1.0 × 10^−^^19^)”, “cell division (*p* = 1.7 × 10^−^^13^)”, and “cellular metabolic process (*p* = 5.9 × 10^−^^7^)”. Branch 3 was not defined due to its small proportion (less than 0.4%). The expression of representative branch markers was superimposed on the cell trajectory plots (Figure 3E).

### 3.4. Characterization of Branch-Specific Markers

We further refined the list of branch-specific markers by integrating bulk RNA-seq data and the Wx algorithm [31]. Only the branch markers upregulated (fold change > 1.5) in THY1 + CAFs compared with that in total CAFs were selected as Branch 1- or Branch 2-specific markers (Figure 4A), since THY1 + CAFs were enriched in Branches 1 and 2, as mentioned above (Figure 3B). Conversely, the branch markers downregulated in THY1 + CAFs were selected as Branch 4- or Branch 5-specific markers. We then determined the mRNA expression of branch markers in total and THY1 + CAFs via qRT-PCR (Figure 4A). As neither total nor THY1 + CAFs were exclusively distributed in each branch, the expression pattern of branch markers did not show any notable trends. However, most Branch 2 markers (70.7%) were upregulated, while most Branch 5 markers (80.0%) were downregulated in THY1 + CAFs. Kaplan–Meier survival analysis of lung adenocarcinoma data from The Cancer Genome Atlas (TCGA) indicated that gene signatures with Branch 2 and Branch 5 markers were associated with poor patient prognosis (Figure 4B). Through the use of scRNA-seq, antigen-presenting CAFs were recently identified as one of the CAF subpopulations in pancreatic cancer [21], and they were shown to suppress the cytotoxic capacity of CD8+ T cells, thus protecting cancer cells [34]. Therefore, we focused on Branch 2, neoantigen-presenting CAFs. In TCGA data, the significant upregulation of an 11-gene signature involving Branch 2 markers in THY1 + CAFs (fold change > 2, *p*-value < 0.05) was associated with worse overall survival of patients with lung adenocarcinoma (Figure 4C), implying that Branch 2 neoantigen-presenting CAFs might promote lung cancer progression. Among 11 Branch 2 markers, five genes (*UBE2T*, *TK1*, *CXCL12*, *KPNA2*, and *HMGB3*) were highly expressed in murine CAFs relative to normal lung fibroblasts (LFs; Figure 4D). Further, high expression levels of *UBE2T*, *TK1*, and *KPNA2* were associated with poor survival rates of patients with lung adenocarcinoma in TCGA data (Figure 4E).

To test the effects of *UBE2T*, *TK1*, *CXCL12*, and *KPNA2* on cancer cell invasion, we knocked down these genes in murine CAFs via siRNAs (Figure 5A and Appendix A). Spheroids composed of 344SQ murine lung cancer cells with either control or *Ube2t*-, *Tk1*-, *Cxcl12*-, and *Kpna2*-knockdown CAFs were seeded into collagen gel, and spheroid invasion was then monitored for 2 days. Eminently, the 3-D invasion of spheroids was suppressed via *Ube2t* or *Kpna2* knockdown but was hardly affected by either *Tk1* or *Cxcl12* knockdown (Figure 5B and Appendix A). Especially, *KPNA2* has been reported to be associated with cancer cell motility and patient survival in breast cancer and non-small-cell lung cancer [35,36]; therefore, we focused on *KPNA2* for the further studies. *Kpna2*-overexpressing LFs promoted spheroid invasion to a greater extent than control fibroblasts (Figure 5C,D). When treated with cisplatin and 5-fluorouracil (5-FU), 344SQ cells cocultured with *Kpna2*-overexpressing LFs were more resistant to these anticancer drugs than those cocultured with control LFs (Appendix A), suggesting that *KPNA2*-high CAFs endow cancer cells with drug resistance. Regarding the macrophage differentiation in the tumor microenvironment, conditioned media from *Kpna2*-overexpressing LFs showed little or no differential effect in inducing polarization of Raw264.7 murine macrophages compared with conditioned media from control LFs (Appendix A). In addition, we engrafted 344SQ cells with *Kpna2*-overexpressing LFs or control LFs into the left lung parenchyma of syngeneic mice (129/Sv). In this orthotopic lung cancer model, *Kpna2*-overexpressing LFs facilitated tumorigenesis and metastasis of 344SQ lung cancer cells to mediastinal lymph nodes and the right lung lobes (Figure 5E).

In order to confirm KPNA2 expression in CAFs, immunohistochemistry was performed on the lung adenocarcinoma tumor sample (Figure 5F). KPNA2 expression was detected in the tumor cells and peritumoral fibroblasts, specifically in the nuclei of fibroblasts. As KPNA2 was also expressed in the tumor cells, we sought to explore the role of KPNA2 in lung cancer cells. Similar to the results in CAFs, *KPNA2* knockdown by shRNAs in human lung cancer cells (A549 and HCC827) inhibited cancer cell migration in Transwell migration assay (Figure 5G and Appendix A) and cancer cell invasion in spheroid invasion assay (Appendix A). *KPNA2* knockdown also suppressed cancer cell migration toward CAFs at the bottom wells (Appendix A). Interestingly, fewer CAFs migrated toward *KPNA2*-knockdown cancer cells than toward control cancer cells in the coculture system (Figure 5H and Appendix A), showing that KPNA2 bidirectionally regulates the interaction between cancer cells and CAFs. Moreover, mRNA levels of *KPNA2* and *UBE2T*, *TK1*, and *CXCL12* were measured in surgically resected tumors from patients with lung adenocarcinoma (*n* = 48), and high *KPNA2, UBE2T, TK1,* and *CXCL12* mRNA levels were associated with poor disease-free survival rates (Figure 5I and Appendix A), implying that *KPNA2* and other Branch 2-specific markers can promote CAF invasiveness and facilitate lung cancer progression.

## 4. Discussion

The interplay between cancer cells and CAFs within the tumor microenvironment regulates cancer progression and metastasis. Depending on the cellular context or external cues, CAFs either facilitate or hinder cancer cell motility and invasiveness [37,38]. In this study, we revealed the heterogeneity of lung CAFs by utilizing scRNA-seq and subsequent bioinformatics analyses. Clustering with UMAP and t-SNE algorithms grouped single CAF cells into several clusters with distinct transcriptomic patterns, yet with a similar distribution tendency between total and THY1 + CAFs. Pseudotime trajectory analysis revealed four major functionally distinct CAF branches (lineages), namely, immunosuppressive, neoantigen presenting, myofibroblastic, and proliferative CAFs (Figure 6). Branch-specific markers were also deduced through trajectory analysis, and some of the markers for neoantigen-presenting CAFs (Branch 2), *UBE2T* and *KPNA2*, were confirmed to promote CAF invasiveness.

In most previous studies on the scRNA-seq analysis of CAFs, tumor-derived cells were pooled together and loaded onto the sequencing platform. Pooled scRNA-seq data were then separated and analyzed via bioinformatics methods for the extraction of CAF-originated data [16,19]. In the present study, we first isolated the CAF population from a lung tumor using cell-type-specific markers prior to sequencing in order to specifically obtain and analyze CAFs. Through this approach, we ensured more accurate and reliable CAF scRNA-seq data without interference or contamination from other types of cells. Nevertheless, it is possible that a small fraction of CAFs was lost during the isolation process, and thus their corresponding scRNA data may have been omitted.

Several markers have been utilized for the isolation of CAFs from solid tumors, including CD44, α-smooth muscle actin (α-SMA, encoded by *ACTA2*), fibroblast-activated protein (FAP), fibroblast-specific protein-1 (FSP-1, encoded by *S100A4*), and THY1 (encoded by *CD90*) [39,40,41]. Here, we used anti-THY1 antibodies to separate the active CAF subpopulation from a surgically resected lung adenocarcinoma tumor. THY1 + CAFs promoted cancer cell invasion and migration in a murine model of lung adenocarcinoma [5,42], localizing in the tumor periphery near actively invading tumor cells [41]. Furthermore, a gene expression signature inferred from THY1 + CAFs was associated with a worse prognosis for lung adenocarcinoma patients, strongly suggesting THY1 as one of the most suitable markers for CAF identification. To compensate for the bias towards the THY1-expressing CAF population, total CAFs were isolated only through negative selection.

Total CAFs and THY1 + CAFs were not clearly distinguished in terms of cellular phenotype or marker expression. Unsupervised clustering analysis of scRNA-seq data revealed that total CAFs and THY1 + CAFs showed similar distribution tendencies for each cluster. However, cell trajectory analysis revealed a prominent distinction between total CAFs and THY1 + CAFs; that is, THY1 + CAFs were predominantly located in Branches 1 and 2, while total CAFs were dominant in Branches 4 and 5. This is because trajectory analysis describes a continuous process (e.g., cell differentiation) that cannot be distinguished through unsupervised clustering methods [43]. Therefore, the current result suggests that THY1 + CAFs have different characteristics from total CAFs even though they originated from the same tumor and around 10% of total CAFs were classified or identified as THY1 + CAF population with THY1 protein expression in the plasma membrane (unpublished data).

Branch 1 deduced from the cell trajectory analysis contained CAFs exhibiting an upregulation of genes involved in apoptosis and the inflammatory response. The apoptosis of stromal cells, including CAFs, can affect the invasiveness and motility of neighboring cancer and immune cells through the release of apoptotic bodies or other factors [44]. Phagocytic engulfment of apoptotic bodies leads to the reprogramming of macrophages into an immunosuppressive phenotype that suppresses T cell proliferation through the upregulation of COX2/PGE2 [45]. Therefore, we named Branch 1 CAFs as “immunosuppressive CAFs”. Among the Branch 1-specific markers were *IGFBP6*, *IFITM3*, and *LGALS3*. IGFBP6 is a potent apoptosis inducer in non-small-cell lung cancer cells and glioma cells [46,47], which has been shown to attract immune cells to the tumor microenvironment [48]. IFITM3 promotes cancer cell proliferation, migration, and metastasis through the TGFβ signaling pathway in prostate cancer cells [49] and facilitates type 2 helper T cell differentiation [50] as well as tumor-associated antigen presentation [51]. Galectin 3 (encoded by *LGALS3*) promotes the interaction between cancer and stromal cells through the secretion of proinflammatory cytokines and the activation of integrin signaling [52], also suppressing immune surveillance via the regulation of T cell and natural killer cell functions [53]. Through these mechanisms, immunosuppressive CAFs can modulate the immune microenvironment and facilitate lung cancer progression.

GO analysis of the upregulated genes in Branch 2 indicated “antigen processing and presentation” as one of the significantly enriched terms. Antigen-presenting CAFs have already been reported as a CAF subpopulation in pancreatic ductal adenocarcinoma [21]. While this CAF subpopulation might process and present antigens via MHC II molecules, a deficiency of costimulatory signals could result in CD4+ T cell deactivation through anergy or Treg differentiation [17,21]. Thus, “neoantigen-presenting CAFs” might function as nonspecialized antigen-presenting cells, which inhibit the optimal T cell response and suppress antitumor immunity within the tumor microenvironment.

Myofibroblastic CAFs remodel the ECM and secrete cytokines, such as CCL2 and TGFβ, leading to fibrosis, immunomodulation, and metastatic progression [54]. Myofibroblastic CAFs have been identified as the main subpopulation in scRNA-seq studies on breast [18], pancreatic [55], bladder [16], and lung cancer [56]. Further, they are associated with resistance to cancer immunotherapy [18,19,55]. In this study, CAFs located in Branch 4 were also named “myofibroblastic CAFs” because the upregulated genes in this CAF population were associated with GO terms, such as “ECM organization”, “wound healing”, “cell motility”, and “cell–matrix adhesion”, which are hallmarks of activated myofibroblasts [57]. Further research is necessary to clarify why myofibroblastic CAFs are more abundant in total CAFs than in THY1 + CAFs and whether these CAFs have any effects on cancer progression and drug resistance.

The majority of total CAFs (77.2%) belonged to Branch 5. Upregulated genes in these CAFs were related to cellular proliferation-associated GO terms, such as “mitotic cell cycle”, “cellular metabolic process”, and “mitochondrion organization”. “Proliferative CAFs” in Branch 5 specifically expressed PRC1, a mitotic spindle-associated protein, and Aurora A kinase (encoded by *AURKA*), both of which are known to promote cancer cell proliferation and tumor growth [58,59]. In addition, proliferative CAFs were previously reported to secrete FGF1, which promotes ovarian cancer cell proliferation and invasion [60], implying that Branch 5 CAFs may promote tumor progression via similar mechanisms.

Branch-specific markers were selected through GO term analysis and Wx, a neural-network-based feature selection algorithm [31]. For Branch 2, 41 marker genes, whose expression levels were associated with poor prognosis in patients with lung adenocarcinoma, were identified. Only 11 genes exhibited similar prognostic value and were highly expressed in THY1 + CAFs relative to total CAFs. Among these, *Kpna2* expression was higher in murine CAFs than in normal lung fibroblasts and was indispensable for CAF-promoted cancer cell invasion. *KPNA2* encodes karyopherin α2 (importin α1), which is involved in nucleocytoplasmic protein transport and promotes proliferation, migration, and invasion in various types of cancer [61]. For example, upregulated karyopherin α2 causes aberrant localization of DNA damage response proteins and worsens the survival rate of patients with breast cancer [35]. KPNA2 was upregulated in cancer tissue and serum from non-small-cell lung cancer patients, enhancing the viability and motility of lung cancer cells [36]. Here, we demonstrated the cancer-promoting effects of KPNA2 in CAFs, suggesting that KPNA2 is closely involved in the interaction between CAFs and cancer cells via paracrine or autocrine pathways.

Previous omics studies, which have analyzed the overall tendency of bulky tumor genomes at the population level, have a limitation in that they cannot provide answers to the single-cell-level questions about intratumoral heterogeneity. The heterogeneity between various cell types within a tumor can be confirmed through scRNA-seq analysis, which can serve as a basis for resolving anticancer drug resistance. The resistance of cancer cells to anticancer drugs, such as apoptosis inducers and angiogenesis inhibitors, is largely facilitated through the interaction with CAFs. This study reveals that a specific subpopulation of CAFs confers anticancer drug resistance to cancer cells, suggesting the possibility of developing new anticancer drugs targeting specific markers of the CAF subpopulation, such as KPNA2. These CAF-specific markers can be used as targets for companion diagnosis predicting the therapeutic efficacy of existing anticancer drugs.

Clearly, this study also has limitations. Unlike most other studies on CAF heterogeneity in which whole tumors were used for scRNA-seq analysis, we isolated CAFs from tumors prior to scRNA-seq; therefore, the characteristic differentiation of CAFs resulting from interactions with other cell types in the tumor microenvironment might not be properly reflected in the analysis. To supplement this point, we tried to verify the analysis results through the cocultures of CAFs with cancer cells and macrophages in vitro. In addition, further study is needed to isolate the CAF subpopulations revealed in this study from lung cancer tissues through flow sorting using cell surface markers.

## 5. Conclusions

Collectively, we identified and characterized novel subpopulations of lung CAFs through scRNA-seq and subsequent comprehensive analyses, presenting several markers specific to each subpopulation. These markers may represent therapeutic targets for the control of CAFs within the tumor microenvironment of patients with lung cancer.

## Figures and Tables

**Figure 1 cancers-14-03486-f001:**
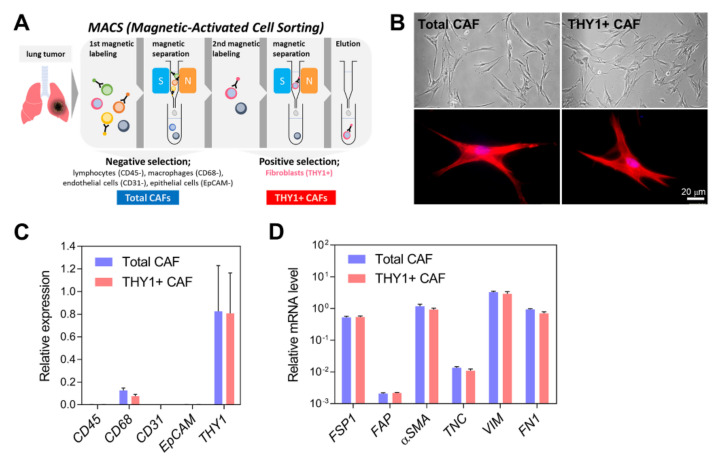
Isolation and characterization of CAFs. (**A**) Total CAFs and THY1 + CAFs were isolated from lung tumors via magnetic-activated cell sorting (MACS). Total CAFs were collected after negative selection using antibodies against CD45 (a lymphocyte marker), CD68 (a macrophage marker), CD31 (an endothelial cell marker), and Ep-CAM (an epithelial cell marker). THY1 + CAFs were then isolated using anti-THY1 antibodies. (**B**) Cell morphology of isolated CAFs. Representative phase-contrast (upper) and fluorescence microscopic images (bottom) are shown. Total and THY1 + CAFs were stained with phalloidin conjugated to Alexa 594 (red) and DAPI (blue). (**C**) qRT-PCR analysis of cell-type-specific markers in total and THY1 + CAFs. Expression levels were normalized to the *RPL32* mRNA level. Data are presented as the mean + SD (*n* = 3). (**D**) qRT-PCR analysis of CAF-specific markers in total and THY1 + CAFs. Data are presented as the mean + SD (*n* = 3).

**Figure 2 cancers-14-03486-f002:**
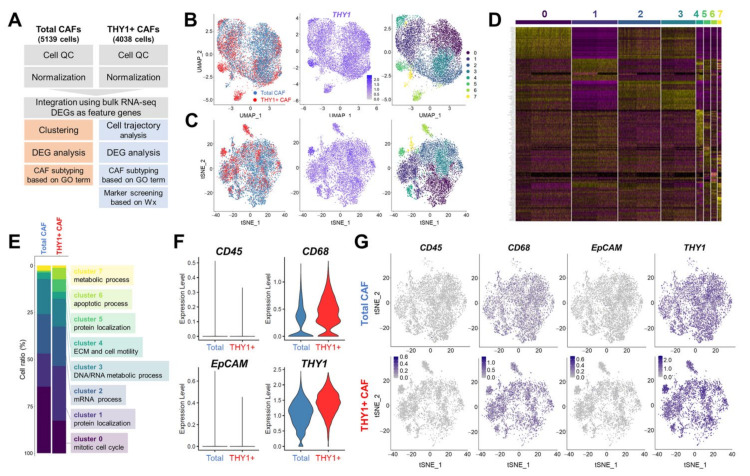
Single-cell RNA sequencing (scRNA-seq) of total and THY1 + CAFs. (**A**) Diagram showing the overall process from CAF isolation to scRNA-seq analysis. (**B**,**C**) Clustering of CAFs via the Seurat method. Data were visualized with UMAP (**B**) or t-SNE (**C**). The distribution of total and THY1 + CAFs (**left**), THY1 expression level (**middle**), and CAF subpopulations (**right**) are presented. A total of eight subpopulations were identified. (**D**) Heatmap of cluster-specific marker genes. (**E**) Gene Ontology (GO) analysis of cluster-specific marker genes. Ratios (%) of each cluster in total and THY1 + CAFs are shown in the bar chart. (**F**) Violin plots showing the expression levels of cell-type-specific markers in total and THY1 + CAFs. (**G**) t-SNE plots showing the expression patterns of cell-type-specific markers in total and THY1 + CAFs.

**Figure 3 cancers-14-03486-f003:**
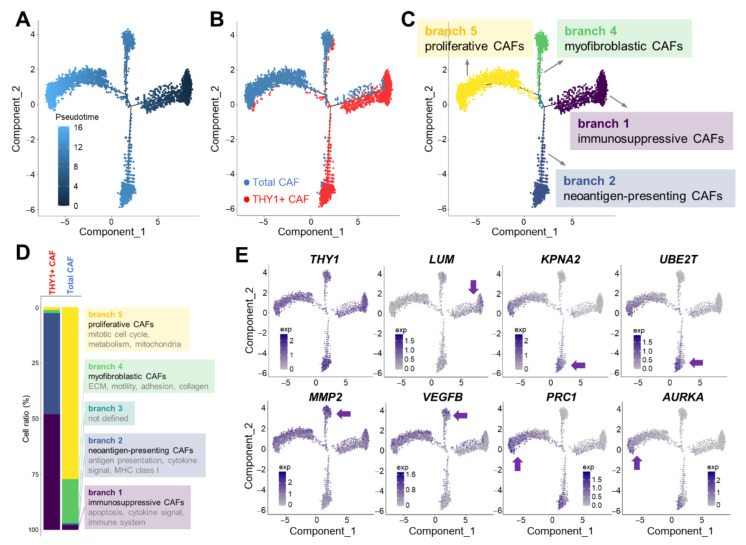
Cell trajectory analysis of CAF scRNA-seq data. (**A**) Cell trajectory analysis of both CAFs. Color intensity means pseudotime. (**B**) Cell trajectory mapping of total and THY1 + CAFs. (**C**) Five branches were identified through the cell trajectory analysis. Branch 1 represents immunosuppressive, Branch 2 represents neoantigen presenting, Branch 4 represents myofibroblastic, and Branch 5 represents proliferative CAFs. (**D**) GO analysis of branch-specific marker genes. Ratios (%) of each branch in total and THY1 + CAFs are shown in the bar chart. (**E**) Cell trajectory plots showing the expression patterns of THY1 and branch-specific markers. The branch corresponding to each marker is indicated by an arrow.

**Figure 4 cancers-14-03486-f004:**
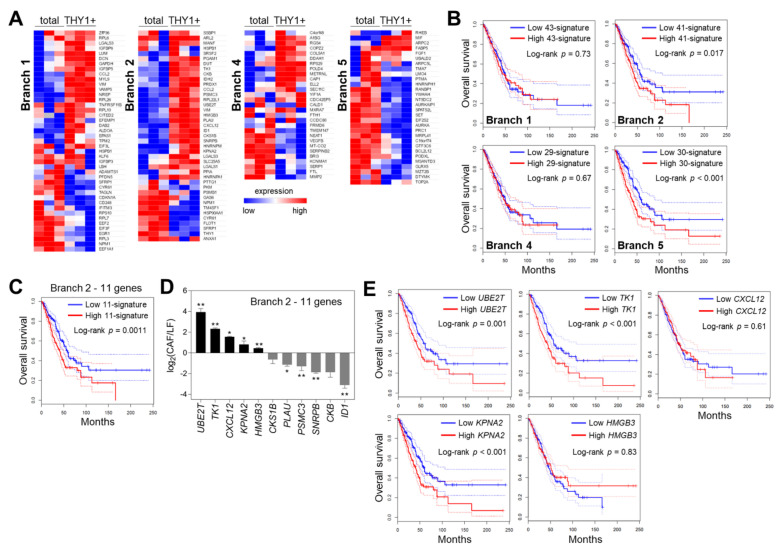
Characterization of branch-specific markers. (**A**) Heatmaps of branch-specific marker expression measured via qRT-PCR in total and THY1 + CAFs. (**B**) Kaplan–Meier plots showing overall survival of lung adenocarcinoma patients (TCGA data, *n* = 478) divided by the median into two groups (high and low) according to the expression levels of branch-specific markers. Dotted lines represent 95% confidence intervals. *p*-Values were determined via the log-rank test. (**C**) A Kaplan–Meier plot showing the effect of 11 signature genes (*UBE2T, TK1, CXCL12, KPNA2, HMGB3, CKS1B, PLAU, PSMC3, SNRPB, CKB*, and *ID1*) from Branch 2 on the overall survival of lung adenocarcinoma patients (TCGA). (**D**) qRT-PCR analysis of 11 genes from Branch 2 in mouse normal lung fibroblasts (LFs) and CAFs. Data are presented as the mean + SD (*n* = 3). * *p* < 0.05, ** *p* < 0.01; two-tailed Student’s *t*-test. (**E**) Kaplan–Meier plots showing the effects of *UBE2T*, *TK1*, *CXCL12*, *KPNA2,* and *HMGB3* expression on the overall survival of lung adenocarcinoma patients (TCGA).

**Figure 5 cancers-14-03486-f005:**
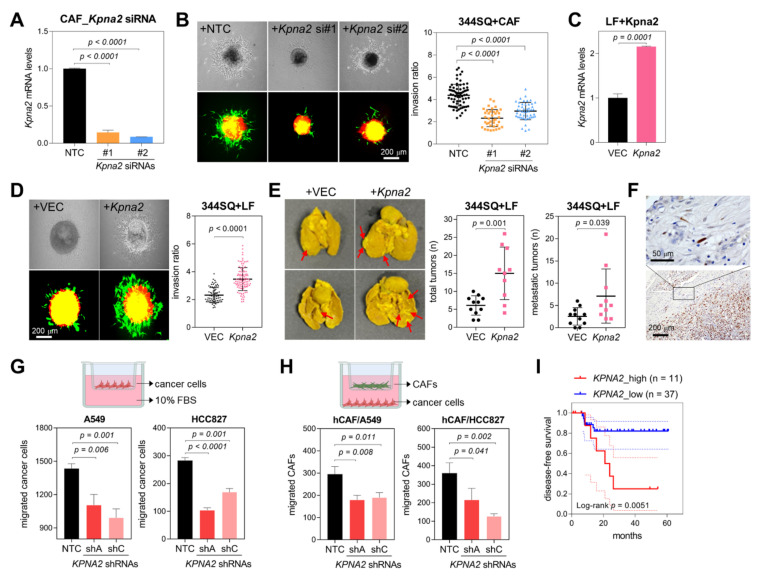
Effects of KPNA2 on CAF activity. (**A**) qRT-PCR analysis of *Kpna2* expression in murine CAFs transfected with nontargeting control (NTC) or *Kpna2* siRNAs (#1 and #2). Expression levels were normalized to the *Rpl32* mRNA level, and values relative to that of the NTCs (set at 1.0) are presented. *p*, two-tailed Student’s *t*-test. Data are presented as the mean + SD (*n* = 3). (**B**) Spheroid invasion assay in 344SQ cells cocultured with CAFs transfected with NTC or *Kpna2* siRNAs (si#1 and #2). These 344SQ cells were labeled with mCherry (red fluorescence), and CAFs were labeled with GFP (green fluorescence). Spheroids made from hanging-drop cultures were seeded on collagen gels and cultured for 2 days. Spheroid invasion ratios (ratio of whole-cell area to central spheroid area) were measured using ImageJ. Mean ± SD (+NTC, *n* = 71; +*Kpna2* si#1, *n* = 38; si#2, *n* = 48). *p*, two-tailed Student’s *t*-test. (**C**) qRT-PCR of *Kpna2* in murine normal lung fibroblasts (LFs) transfected with an empty vector (+VEC) or a *Kpna2*-overexpressing vector (+*Kpna2*). *p*, two-tailed Student’s *t*-test. Data are presented as the mean + SD (*n* = 3). (**D**) Spheroid invasion assay in 344SQ cells cocultured with LFs transfected with control or *Kpna2*-overexpressing vector. Mean ± SD (+VEC, *n* = 72; +*Kpna2*, *n* = 156). *p*, two-tailed Student’s *t*-test. These results are representative of at least three separate experiments. (**E**) Orthotopic injection of 344SQ cells (5 × 10^4^ cells/mouse) with LF + VEC or LF + *Kpna2* (5 × 10^4^ cells/mouse) into syngeneic mice. After 11 days, mice were necropsied, and the lungs were stained with Bouin’s solution to visualize tumor nodules (red arrows, left panels). Graphs show total and metastatic tumor nodules. Mean ± SD (+VEC, *n* = 11; +*Kpna2*, *n* = 10). *p*, two-tailed Student’s *t*-test. (**F**) Immunohistochemical staining of KPNA2 in lung adenocarcinoma tumor sections. The high-magnification image (upper) shows nuclear staining of KPNA2 in fibroblasts. (**G**) Transwell migration assay in human lung cancer cells (A549 and HCC827) transduced with control (NTC) or *KPNA2* shRNAs (shA and shC). Cancer cells were seeded in the inserts, and complete medium with 10% FBS was added to the bottom wells. After 24 h, migrated cancer cells were photographed and counted. Mean + SD (*n* = 3). *p*, two-tailed Student’s *t*-test. (**H**) Transwell migration assay in human CAFs cocultured with *KPNA2*-knockdown (shA and shC) or control (NTC) lung cancer cells. CAFs were seeded in the inserts, and cancer cells were seeded in the bottom wells. After 24 h, migrated CAFs were photographed and counted. Mean + SD (*n* = 3). *p*, two-tailed Student’s *t*-test. (**I**) A Kaplan–Meier plot showing the disease-free survival of patients with lung adenocarcinoma. Patients were divided into two groups (*KPNA2*_high and *KPNA2*_low) based on their *KPNA2* mRNA expression levels. *p*-Value was determined via the log-rank test.

**Figure 6 cancers-14-03486-f006:**
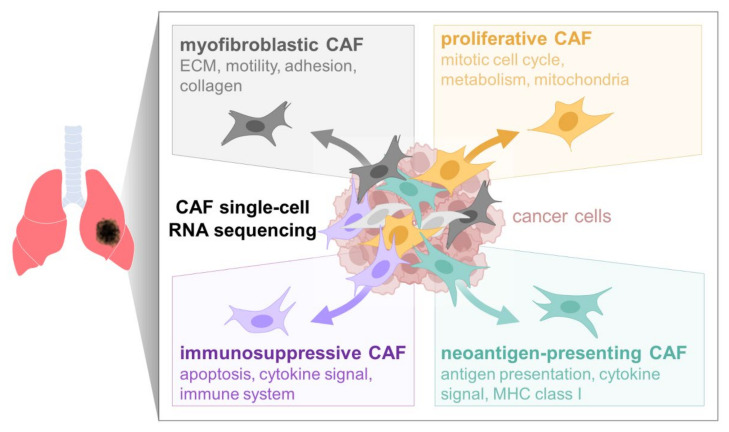
Subpopulations of lung CAFs identified in this study though scRNA-seq.

## Data Availability

The data presented in this study are available on request from the corresponding authors.

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
