# Peer review of "Identification and Characterization of Cancer-Associated Fibroblast Subpopulations in Lung Adenocarcinoma"

_cancers, 2022, doi:10.3390/cancers14143486_

Round 1

Reviewer 1 Report

In this manuscript, Kim et al. use scRNA-Seq to identify subpopulations if cancer-associated fibroblasts in lung adenocarcinoma. While the significance and presentation are very high, there are some minor issues which, once clarified, would improve the manuscript.

  1. THY1 was used to isolate a subpopulation of CAFs that is largely representative for all CAFs, including high THY1 expression in 'total CAF' population (Figure 2G). However, in the discussion the authors mention that "around 10% of total CAFs were THY1+ (unpublished data)" (line 413). This statement is confusing considering the data presented in the manuscript. Please clarify how this estimate was obtained and what can be the reason for these discrepancies.
  2. In several figures the number of replicates is shown as "(n=3)". Are these technical or biological replicates?
  3. In Figure 3E, the gene names should be in italics
  4. In Figure 4, the authors focused on KPNA2 based on its enrichment in CAFs over normal fibroblasts. However, 3 other genes (UBE2T, TK1 and CXCL12) show even higher enrichment in CAFs compared to normal fibroblasts. These 3 genes, and preferably all 11 'signature' genes, should be analyzed the same way as KPNA2 in Figure 4E.
  5. Figure 5E: the scale bar is missing

Reviewer 2 Report

·         As the authors mention, CAFs are important contributors and regulators of the tumor-stroma cross talk during disease progression. This includes promoting tumor proliferation and stemness, aside from enhancing tumor invasiveness/motility, as well as inducing an anti-inflammatory, tumor-supportive immune response. Hence, it would be relevant to add extra experiments to the co-culturing assays, in order to establish the role of CAF-expressed KPNA on the tumor proliferation status, DNA damage, as well as the macrophage phenotype.

·         As KPNA2 is mainly expressed in the tumorous tissue, and not the stromal compartments of the lungs in lung cancer, it would perhaps be (more, but also) physiologically relevant to inhibit KPNA in cancer cells in functional assays, as well as in co-culturing assays with CAFs (and perhaps macrophages) and assess the stromal cell phenotype.

·         The readers could benefit from a graphical representation of the main results of the study, in terms of a visual representation of the main identified branches of CAFs and their characteristics.

·         The authors should reflect more on the therapeutic potential/translational relevancy of these findings.

Reviewer 3 Report

In this paper, Kim D et al have identified a novel subpopulation of lung carcinoma infiltrating CAF, highlighting its potential value as a therapeutic target.

The article is well written, it is clear and sound well organized; the authors have carried out numerous experiments with scientific criteria and appropriateness, the material and method section describe appropriately the techniques used, and the figures are well represented and well described in the legend.

Reviewer 4 Report

This is an interesting paper that aimed to identify subpopulations of cancer-associated fibroblast subpopulations isolated from human lung adenocarcinomas and describe their transcriptomic as well as functional characteristics through single-cell RNA sequencing (scRNA-seq) and subsequent bioinformatics analyses. Overall the study is well written and English language is adequate. Strengths and limitations of the study should be stated more clearly. I would also suggest to include further discussion on the future direction and potential clinical application of the results.
